# The Prime and Integral Cause of Cancer in the Post-Warburg Era

**DOI:** 10.3390/cancers15020540

**Published:** 2023-01-16

**Authors:** Salvador Harguindey, Stephan J. Reshkin, Khalid O. Alfarouk

**Affiliations:** 1Institute of Clinical Biology and Metabolism (ICBM), 01004 Vitoria, Spain; 2Department of Biosciences, Biotechnology and Biopharmaceutics, University of Bari, 70125 Bari, Italy; 3Zamzam Research Center, Zamzam University College, Khartoum 11123, Sudan

**Keywords:** pH in cancer primal etiology, Warburg effect nowadays, changing hallmarks in cancer, historical mishaps in metabolic cancer research, integrations among orthodox and heterodox oncology, pH-related therapeutic implications

## Abstract

**Simple Summary:**

In this perspective, we have gone back to the beginnings of metabolic cancer research to grow from its roots. From the time of Otto Warburg, the field of cancer research has progressively learned the fundamental importance of metabolism. In this contribution, we also clarify the errors committed along the way to reaching the present state of the art. Nowadays, a multitude of intermediary causes of cancer have been identified that are shown to act through final and integral causes. This new conceptualization allows a better understanding of the mechanisms working behind the famous Warburg effect. At the same time that the initial limitations and misinterpretation of this effect are considered, we also explain its origin through the lens of the pH-related cancer paradigm. Here it is also shown that nowadays this hydrogen ion (H^+^)-mediated perspective is key to understanding the role of the pH-approach as the prime and *sine que non* metabolic cause of cancer.

**Abstract:**

**Back to beginnings.** A century ago, Otto Warburg published that aerobic glycolysis and the respiratory impairment of cells were the prime cause of cancer, a phenomenon that since then has been known as “the Warburg effect”. In his early studies, Warburg looked at the effects of hydrogen ions (H^+^), on glycolysis in anaerobic conditions, as well as of bicarbonate and glucose. He found that gassing with CO_2_ led to the acidification of the solutions, resulting in decreased rates of glycolysis. It appears that Warburg first interpreted the role of pH on glycolysis as a secondary phenomenon, a side effect that was there just to compensate for the effect of bicarbonate. However, later on, while talking about glycolysis in a seminar at the Rockefeller Foundation, he said: “Special attention should be drawn to the remarkable influence of the bicarbonate…”. Departing from the very beginnings of this metabolic cancer research in the 1920s, our perspective advances an analytic as well as the synthetic approach to the new “pH-related paradigm of cancer”, while at the same time addressing the most fundamental and recent changing concepts in cancer metabolic etiology and its potential therapeutic implications.

## 1. Introduction: Otto Warburg Today: The pH Factor as the Missing Link

At one point in his research life, Otto Warburg became aware that environmental pH was an important parameter in maintaining glycolysis in his culture systems. In spite of this, it seems that he was never fully aware of the fundamental importance of acid-base conditions on cancer intermediate metabolism. At least, he did not address the subject of the role of pH in glycolysis again in his later works and talks [1,2,3] (see Table 1 for a summary of the main historical hallmarks of metabolic cancer research). Furthermore, taking into account that during Warburg’s lifetime there were no good methodologies to determine intracellular pH (pHi), he probably took for granted that the cytoplasm of cancer cells was acidic because of their high production of lactic acid. However, nowadays we know that the real situation is exactly the opposite and that cancer cells of all malignant tumors and leukemias are in a certain state of metabolic alkalosis, sometimes so severe that it is barely compatible with life [1,2,4]. Today we also know that among the many allosteric factors that affect glycolysis, in either normoxia or under hypoxic conditions (hormones, ions, viruses, physicochemical drugs, genes, oncogenes, metabolic factors and therapeutic drugs (see Table 2 and ref. [5]), the H^+^ concentration is the determinant and common factor in the control of glycolysis.

## 2. Why Warburg Was Right and Wrong at the Same Time

In 1956, very heated disagreements arose, mainly between Otto Warburg and Sidney Weinhouse, while discussing the meaning of glycolysis and the respiratory impairment of malignant cells in cancer and their relationship to oxidative phosphorylation [1,6]. Unfortunately, the pH/glycolysis relations in cancer were completely absent from those passionate discussions. Furthermore, we can now realize that missing the pH relationship helped to create a great deal of confusion during the next decades until the present times. During all those years many different theories were proposed to clarify the true role of the increased glycolysis of tumors [7]. Nowadays, these historical limitations and clashes can be better understood only after the cause-effect relationships of pHi elevations in upregulating glycolysis began to be considered in the late 1960s and early 1970s.

Regrettably, this attention to the cause-effect of elevations of intracellular pH on glycolytic stimulation started during Warburg’s old age (he died in 1970 at the age of 86 years) and its interest was mainly outside the cancer context [8,9,10,11,12,13,14,15,16]. Thus, Warburg did not really have a chance to better interpret either the etiopathogenic relationships between pHi and aerobic and/or anaerobic glycolysis; much less on the role of the stimulatory and carcinogenic effects of growth factors (GFs), viruses, oncogenes, membrane-bound proton transporters (PTS), HIF and many other compounds of different origins and natures. All of which raises pH and stimulate glycolysis and cancer metabolism (Table 2). These relationships were discovered during and after the 1980s, well after Warburg’s death [12,14,17,18,19,20,21,22,23,24,25,26,27].

Since then, interest has been steadily growing concerning the intimate and interdependent relationships between pHi/pHe and the complex H^+^-dynamics of cancer; namely, the highly pathological intracellular alkalization/extracellular acidification of tumors known as cancer proton reversal (CPR), as well as the concerted strategy of cancer cells and tissues as we know it today [5,28,29,30] (Figure 1).

The above-considered historical reasons fully justify the fact that Warburg could not possibly be right on what he considered to be “the prime cause of cancer”, namely the aerobic glycolysis and the impairment of respiration of tumors, something that he defended until his death. However, we should now admit that he was very close to it [31]. This is because, as is becoming accepted by different research groups, the prime cause of cancer and/or of cellular malignant transformation is also the main factor working behind the induction of aerobic glycolysis itself, namely, the selective intracellular alkalization of all tumors and leukemias [32]. Furthermore, even the normal functioning of the tricarboxylic cycle (CTC) was shown to be hindered by alkalosis [13].

Finally, recent publications from three different laboratories have strongly defended that the Warburg effect can be fully explained by a selective increase in the pHi of cells and the upregulating effects of this intracellular alkalization on the activation of aerobic glycolysis [28,33,34,35]. For a more complete explanation of this phenomenon, see ref [5]. Recently, it has been demonstrated that any system that increases the export of protons or proton equivalents from a cell drives the Warburg Effect (the upregulation of glycolytic enzymes) with concomitant increases of both in vitro and in vivo aggressive behaviors [36].

## 3. From the Initial pH-Glycolysis Associations to the Present State of the Art in Metabolic Cancer Research

Some seminal clinical insights of acid-base abnormalities in both the onset and regressions of cancer were published as soon as 1975 [37,38]. A few years later, we insisted on the fact that alkalization was fundamental in inducing cancer [39,40]. Since then, the high pH-related paradigm on the etiopathogenesis of malignancy, as well as its stimulatory effects on cancer growth has gained increasing attention [5,28,30,41,42,43,44,45,46,47,48,49,50,51,52,53]. Today, the pH-centric integral perspective of cancer is finally recognized by growing sections of the research community in basic, preclinical and clinical oncology and constitutes a brand-new paradigm in medicine.

After all, proton export alone is sufficient to induce cell derangements, from dysplasia to full malignant transformation in different human tumors and very different situations [42,43,46,47,48,49,50].

## 4. On the Changing Hallmarks of Malignancy in pH-Related Metabolic Cancer Research

The pH-centered metabolic hallmarks were not considered in the already classical hallmarks of cancer described by Hanahan and Weinberg in 2000 [54] and later in 2011 [55]. Nor have they been considered in a more recent publication by Hanahan on cancer hallmarks [56]. However, Kroemer and Pouyssegur, approaching the problem from the pH perspective, have described seven new pH-related metabolic hallmarks of cancer [57]. Other cancer-selective molecular, biochemical and metabolic abnormalities, like the CPR and the defects of mitochondrial oxidation, among other fundamental pH-related factors, have been progressively included as new hallmarks of malignancy [29,30,31,58,59,60,61,62,63]. This new paradigm of the hydrogen ion dynamics of cancer has recently permitted the integration of many etiopathogenic intermediary factors of different origins and natures causing cancer within a single and wider ranged model based upon cellular alkalization, the upregulation of NHE1 and deranged “mass effect” of the concentration of hydrogen ions (H^+^) within the intracellular space [29] (Table 2 and Figure 1).

## 5. Therapeutic Implications: Closing Gaps

The therapeutic utilization of a concerted mixture of different proton transport inhibitors (PTIs) in human cancer was initially proposed more than a decade ago [64]. The aim of this homeostatic and/or acid/base-related therapeutic approach to malignancy is to first control and then selectively decrease the pHi of cancer cells to acidic levels in order to induce their apoptosis (anti-Warburg effect in therapy) [57,64,65]. At the same time, these therapeutic efforts aim to reverse the secondary tumoral environmental-extracellular acidification (TME) characteristic of the CPR of malignant cells and tissues. CPR is also a fundamental factor in the escape of malignant tumors from the organism’s immune defense mechanisms [66,67].

Furthermore, the disruption of cellular and microenvironmental acid-base homeostasis, and the mass effect of its H^+^-dynamics, have also been shown to be essential in the effects of a wide array of chemical carcinogens. This knowledge opens new possibilities in the control of environmental carcinogenesis [68,69] (Table 2). Moreover, in this line, our group has proposed new pH-dependent avenues for the treatment of human malignancies, ranging from brain tumors [29] to breast cancer [51,52] to other malignancies [28,30]. During the last few years, other groups have also activated clinical efforts to exploit the significantly diseased pH-related metabolic aspects of malignant tumors in cancer treatment [70,71].

Most recently, and still within the pH-centric therapeutic paradigm, an increasing number of promising repurposed drugs are being introduced in the pharmacological armamentarium available to clinicians (off-label use). Some of them, such as the beta-blocker propranolol, inhibit NHE1 inducing anti-proliferative and pro-apoptotic effects and also activate an anti-tumor immune response in vivo [72]. Furthermore, melatonin has been shown to act as an anticancer drug in glioblastoma via lowering of the pHi [73], in this way also inhibiting the Warburg effect [74]. Most recently, a highly interesting publication on the anti-Warburg effect as an important therapeutic perspective has been thoroughly reviewed [75].

These recent developments make it even more interesting that an orthodox anticancer drug such as cisplatin (CDDP) also inhibits NHE1. Indeed, the first effect that can be detected after the administration of CDDP is, surprisingly, the induction of cellular acidification via the inhibition of H^+^ efflux in sensitive cancer cells [76]. On the contrary, NHE1 hyperactivity and/or upregulation of this and/or other membrane-bound proton pumps (PP) and transporters (PT) induces not only CDDP resistance by elevating pHi, but also multiple drug resistance (MDR), as it was long ago shown in seminal and similar studies on the parallelism between adriamycin resistance and progressive increases in pHi [77,78,79] (Figure 1). An increasing number of studies also advise the utilization of alkalizing therapy to counter the microenvironmental acidity of tumors and, in this way, improve chemotherapy resistance, anticancer immunity and deactivate the metastatic process [80]. Most recently too, fermented wheat germ extract (FWGE) has been shown to inhibit metastatic tumor dissemination during and after chemotherapy, surgery, and/or radiation in cancer patients. It also improves survival in melanoma patients while being beneficial in the treatment of autoimmune diseases [81]. Indeed, FWGE is becoming a promising agent in modern anticancer therapeutics, since it has been shown to restore mitochondrial function, suppress the Warburg effect and inhibit in vivo tumor growth [82,83,84,85].

All these developments help to consider further efforts towards uniting traditional and orthodox oncology with non-orthodox and less traumatic approaches into one single concept within an all-comprehensive model of biological synthesis and unification.

Finally, the initial pH-related paradigm has been progressively extended beyond its initial cancer context to the pathogenesis of human neurodegenerative diseases (HNDDs) [28], mainly to the etiopathogenesis and metabolic treatment of multiple sclerosis (MS) [32]. Finally, a Warburg-like effect has been recently blamed for aerobic glycolysis-mediated neuronal degeneration in some cases of spontaneous Alzheimer’s disease [86]. The Warburg effect has also been recently implicated, not only in malignancy and neurodegeneration but also in the pathogenesis of both bacterial and viral infections [87]. Indeed, the COVID-19 pandemic has underlined the fact that the Warburg effect is behind the rapid dissemination of other different diseases within a hypoxic environment, for instance, sustaining the glycolysis of infected lung endothelial cells [87]. Moreover, immunity has now been found to be glycolysis-dependent for its proper functioning [87,88]. However, while the full understanding of the mechanisms by which viruses and bacteria activate the Warburg effect still remains elusive, it is rational to think that the same, or very similar, pathological dynamics of the hydrogen ion should be implicated, as has been suggested for a long time [89,90,91,92].

## 6. Conclusions: Towards Biological Unifications

The pH and homeostasis-centered cancer paradigm follow the quest towards unification of scientific visions, and it aims at the reconciliation of apparently unrelated phenomena. After integrating both the old and latest data available into the new, wide-ranged and unitarian pH-centric (or H^+^-dynamics) paradigm, it can be stated beyond doubt that the prime cause of cancer never was the aerobic glycolysis of tumors and/or the respiratory impairment of cancer cells, as Warburg defended all his life, but a final common and *sine qua non* pathway that may now allow a deeper understanding of the most ‘basic’ origin of cancer. The perspective considered here jumps one step ahead to define that the main metabolic factor behind aerobic glycolysis is a selective intracellular alkalization of cells in all tumors, which, in turn, is the prime and ultimate cause of cancer. This integral interpretation finally advances a beneficial side-effect that leads to further unifications in biology. It is concluded that this highly energetic model will be continuously opening new therapeutic avenues to improve the treatment of, at least, human cancer and neurodegenerative diseases.

## Figures and Tables

**Figure 1 cancers-15-00540-f001:**
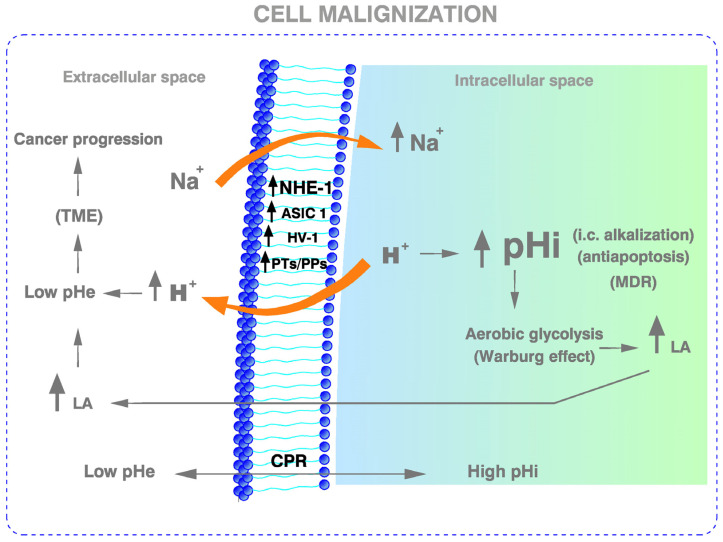
pH-related interplay among the most significant membrane-bound transporters, cellular alkalization, aerobic glycolysis and the Warburg effect. Cell alkalization, as induced by multiple previous upstream processes (see Table 2) is the primal and integral mediator of malignant transformation and the main metabolic and acid-base homeostatic derangement and anti-apoptotic factor, which is also fundamental in MDR. The secondary interstitial (TME) acidification of tumors originates the cancer proton reversal (CPR). Finally, this concatenation of pathological energetic changes drives the cascade of cancer progression and the metastatic process. Abbreviations: NHE1, Na^+^/H^+^ antiporter isoform 1; ASIC1, acid-sensing ion channel type 1a; Hv1, voltage-gated Na+ and H+ channel isoform 1; PTs, proton transporters; PPs, proton pumps; CPR, cancer proton reversal; MDR, multiple drug resistance; LA, lactic acid. For further details, see the text.

**Table 1 cancers-15-00540-t001:** Progressive milestones of metabolic cancer research during the last one hundred years.

1920s. Discovery of the aerobic and fermentative glycolysis of tumors by Otto Warburg seminal and extraordinary discoveries.
1956. Heated discussions, mainly between Otto Warburg and Sidney Weinhouse, with Arthur Schade and Dean Burk as witnesses, on the meaning of tumor cells glycolysis and respiration in cancer etiology.
1960s and 1970s. Discovery of the stimulatory effects on glycolysis and carcinogenesis of different pH-elevating growth factors.
1980s. Discovery of the Na^+^/H^+^ antiporter (NHE) and its inhibition in arresting cancer cell growth.
1980s and 1990s. Increasing description of different membrane-bound hydrogen ion (H^+^) transporters (PTs) as well as a high pHi in cell transformation of fibroblasts.
1990s. First demonstration that certain proton transport (PTs) stimulators induce cell transformation by increasing pHi while inhibiting H+ efflux inhibits cancer growth and facilitates the induction of cancer cell apoptosis.
2000s. (A) Increasing research on the potential importance of proton transport inhibitors (PTI) in cancer treatment and the discovery of the carcinogenic effects of different Na^+^/H^+^ antiporter upregulating factors in cancer etiology, etiopathogenesis, growth and the metastatic process (virus, PTs, chemical carcinogens, etc.) (See Table 2). (B) Description of pH-unrelated hallmarks and new pH-related hallmarks of cancer. (C) First publications of pH-related integral measures as a concerted therapeutic approach to different human malignancies.
2010s. Description of cancer proton reversal (CPR) as a highly selective cancer hallmark of cancer. An increasing recognition of the new pH-centric anticancer paradigm in basic and clinical oncology. Finally, the first explanations of the opposite relations between cancer and human neurodegenerative diseases (HNDDs).
2020s. Extension of the Warburg effect to the pathogenesis of different human pathologies beyond cancer, including human neurodegenerative diseases and infections.

**Table 2 cancers-15-00540-t002:** Intermediary neoplastic drivers become carcinogenic through the final increase in cell pH and/or NHE1 upregulation.

Gene products such as Bcl-2
Virus (e.g., Human Papilloma Virus)
Oncogenes and viral products (e.g., HPV-E7)
Modification/mutations/over-expression in other pH transporters (PTs) and proton pumps (PPs)
Chemical carcinogens (groundwater arsenic salts, polycyclic aromatic hydrocarbons)
Chronic and intermittent hypoxia
Mutations and genomic instability (BRCA1/2)
Aging (which Warburg called “Time Caused Cancer”)
Glucose overload
Mitogens (VEGF isoforms, EGF, interleukin isoforms, TGF isoforms, PDGF isoforms, etc.)
Hormones and cytokines (growth hormone, prolactin, glucocorticoids, etc.)
p53 deficiency/mutations
Immune evasion

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
