# Peer review of "The Prime and Integral Cause of Cancer in the Post-Warburg Era"

_cancers, 2023, doi:10.3390/cancers15020540_

Round 1
Reviewer 1 Report
Hi,
The manuscript appears to contain interesting and valuable information; I have only suggestions for changing "the" to "in" because it is repeated in line 112.
Author Response
The manuscript appears to contain interesting and valuable information; I have only suggestions for changing "the" to "in" because it is repeated in line 112.
Response 1: Our response for Point 1.
Line 112 has been changed to “Why Warburg was right and wrong at the same time”, and “the” has been changed to “at”.
The authors appreciate the high scoring rates of this reviewer.
Reviewer 2 Report
Table 1 and 2. These tables are interesting and well done; however I suggest including reference(s) to each description made in it, for example
"1920´s. Discover...." Ref(s) 13, 14, 15, etc.... also please describe better the asterisk, in which paragraph the readers will be obtain more details.
Please review lines 189-191"Within this vein too..." the redaction is not completely understandable.
Overall comments:
The manuscript is interesting and well done; however, English needs to be reviewed by an expert, tables; should it be modified, according to my suggestions. I recommend make a figure in which explain the Warburg effect into tumor microenvironment, also the effects of treatments with the Warburg effect.
Author Response
Table 1 and 2. These tables are interesting and well done; however, I suggest including reference(s) to each description made in it, for example
"1920´s. Discover...." Ref(s) 13, 14, 15, etc.... also please describe better the asterisk, in which paragraph the readers will be obtain more details.
Response No. 1:
As this is a Perspective rather than a Review, we respectfully request that it not be necessary to include said references to each description.
We decided not to include references in the tables, as we have previously done in some of our previous reviews of this publisher (MDPI), first, for clarity, then to avoid redundancy with some of our previous publications and finally, because the list of references would be around double of the present. We ask this reviewer to consider that all the contents of the tables are already clearly mentioned in the text,
Response No 2.
Please review lines 189-191: "Within this vein too..." the redaction is not completely understandable.
In the new text, “Within this vein too” has been changed to “Also in this line” (lines 175-176)
Overall comments of reviewer No. 2.
The manuscript is interesting and well done; however, English needs to be reviewed by an expert, tables; should it be modified, according to my suggestions. I recommend make a figure in which explain the Warburg effect into tumor microenvironment, also the effects of treatments with the Warburg effect.
Our response:
Major English changes have been introduced through the manuscript.
We have also modified the Tables to fit the format of Cancers and introduced a new figure, entitled: “Cell malignization”; as proposed by the reviewers.
Reviewer 3 Report
The authors discussed the prime cause of metabolic cancer after the Warburg era. This topic is highly interesting to the subject group readers. At the beginning, the authors tried to point out the limitations of Warburg effect and later pointed out other theories on root cause of metabolic cancers. However, based on the discussion, it seems the conclusion made does not completely support the objective of this study, and the claim of root cause of metabolic cancer is not clearly understandable based on the discussed information. Additionally, there are many technical errors in data presentation, particularly tables, some minor typos or duplicate words in sentence. For example, title of section 2. So, this perspective study must need to be more comprehensive regarding information, data presentation and including illustration of mechanisms of metabolic cancer showing the root cause. Adding one table comparing the findings on metabolic cancer prior Warburg era, argument and discussion by Warburg in establishing the theory, and current stand points on metabolic cancer. Finally, support the current standpoint by comprehensive discussion and illustration of root cause.
There are some minor issues like keywords and abstract writing in the manuscript that should be resolved.
Author Response
see cover letter
Reviewer 4 Report
1. We have gone back to the beginnings of metabolic cancer research one hun- 9 dred years ago to develop from its roots the knowledge of its fundamental importance in the field 10 of cancer research. Such long sentences shall be avoided as it disrupts the interest of the readers. The sentences should be crisp and short. Rewrite all the long sentences in a crisp way.
2. At the same time, we attempt to clarify the errors committed along the way to 11 reaching the present state of the art. Grammatical flaw. It seems like there are many grammatical flaws throughout the manuscript. The authors are advised to proofread the manuscript using GRAMMARLY or any other software to rectify all these flaws.
3. A recent article related to Warburg effect was published in highly reputed journal of cancer. The authors can add some relevant information from that too. https://doi.org/10.1016/j.bbcan.2021.188568
4. Tables are completely out of the format.
5. Some figures must be embedded as graphical representation is a must to attract the interest of the audience.
Author Response
Reviewer comment No. 1. We have gone back to the beginnings of metabolic cancer research one hundred years ago to develop from its roots the knowledge of its fundamental importance in the field 10 of cancer research. Such long sentences shall be avoided as it disrupts the interest of the readers. The sentences should be crisp and short. Rewrite all the long sentences in a crisp way.
Our response:
As stated to Reviewers 1 & 2 and to the Editor: We have made major English and style changes throughout the manuscript and shortened many sentences.
Reviewer Comment No. 2
At the same time, attempt to clarify the errors committed along the way to reaching the present state of the art. Grammatical flaw. It seems like there are many grammatical flaws throughout the manuscript. The authors are advised to proofread the manuscript using GRAMMARLY or any other software to rectify all these flaws.
Our response
As just stated above we have done this.
Reviewer comment No. 3. A recent article related to Warburg effect was published in highly reputed journal of cancer. The authors can add some relevant information from that too. https://doi.org/10.1016/j.bbcan.2021.188568
Our response:
Thank you for this suggestion, we have now included this reference.
Reviewer comment No. 4. Tables are completely out of the format.
Our response:
They are now formatted.
Reviewer comment No. 5. Some figures must be embedded as graphical representation is a must to attract the interest of the audience.
Our response
Thank you for the suggestion, we have now included a new figure to better represent the content of this Perspective.
Reviewer 5 Report
In the manuscript entitled “The prime metabolic cause of cancer in the post Warburg era”, Harguindey and co-authors propose to refresh the current knowledge of the tumor microenvironment metabolic features. The manuscript is very well written and gives an important view of the tumor pH paradigm starting from the Warburg perspective to the current efforts in the design of therapeutics based on tumor’s acidic pH. However, it would also be important to include one or two paragraphs explaining with clarity and more detail the state of the art of the TME pH.
Minor comments:
The abstract is too focused on Warburg findings. If the “simple summary” was not present, readers would not understand the objectives of the manuscript.
Chapter 2 has a typo – replace by “Why Warburg was right and wrong at the same time”
Authors refer twice in the text the abbreviation of cisplatin (CDDP). Moreover, cis-platinum should be replaced by cisplatin – the common name given to this chemotherapeutic agent.
Author Response
In the manuscript entitled “The prime metabolic cause of cancer in the post Warburg era”, Harguindey and co-authors propose to refresh the current knowledge of the tumor microenvironment metabolic features. The manuscript is very well written and gives an important view of the tumor pH paradigm starting from the Warburg perspective to the current efforts in the design of therapeutics based on tumor’s acidic pH. However, it would also be important to include one or two paragraphs explaining with clarity and more detail the state of the art of the TME pH.
Minor comments:
The abstract is too focused on Warburg findings. If the “simple summary” was not present, readers would not understand the objectives of the manuscript
Our response:
Thank you for this comment, we have now modified both the Simple summary, the Abstract and some of the text to make the focus of the Perspective clearer.
Chapter 2 has a typo – replace by “Why Warburg was right and wrong at the same time”
Our response:
This typo has been corrected. As stated to Reviewers 1, 2 & 3 and to the Editor, we have made major English and style changes throughout the manuscript to correct that other typos and make the general content much clearer.
Reviewer comment:
Authors refer twice in the text the abbreviation of cisplatin (CDDP). Moreover, cis-platinum should be replaced by cisplatin – the common name given to this chemotherapeutic agent.
Our response
Thank you for this comment and suggestion- we have done so.
Round 2
Reviewer 4 Report
All the comments have been addressed and manuscript can now be accepted for publication.